# Investigating the Psychometric Properties of the Emotion Regulation Flexibility Questionnaire in the Italian Context

**DOI:** 10.3390/ejihpe15080165

**Published:** 2025-08-19

**Authors:** Giada Mignolli, Daiana Colledani, Francesco Tommasi, Anna Maria Meneghini

**Affiliations:** 1Department of Human Sciences, University of Verona, 37129 Verona, Italy; anna.meneghini@univr.it; 2Department of Psychology, Sapienza University of Rome, 00185 Rome, Italy; daiana.colledani@uniroma1.it; 3Department of Social and Political Sciences, University of Milano, 20122 Milano, Italy

**Keywords:** emotion regulation flexibility, scale validation, structural equation modelling, emotion dysregulation, eudaimonic and hedonic well-being

## Abstract

Background: Following the idea that individuals engage with different strategies to regulate their emotional experiences, scholars in the field of psychology have shown increasing interest in the notion of emotion regulation flexibility. Despite growing attention to this construct, validated instruments for assessing individuals’ capacity to choose among different emotion regulation strategies effectively are limited, particularly in non-English-speaking contexts. The present study aims to extend the use of the Emotion Regulation Flexibility Questionnaire by providing a validation of the Italian version and supporting its generalisability. Methods: The Italian Emotion Regulation Flexibility Questionnaire (IT-ERFQ) was included in a cross-sectional study involving *N* = 887 participants (60.4% female). Exploratory and confirmatory factor analyses were conducted, along with tests of measurement invariance across gender and age groups, assessments of internal consistency, and evaluations of external validity. Results: The IT-ERFQ showed a clear unidimensional structure, good internal reliability, and full measurement invariance across groups. The eight-item Italian version correlated negatively with emotion dysregulation and positively with well-being. Conclusions: These findings provide strong evidence for the psychometric soundness of the IT-ERFQ and support its use in both basic and applied research.

## 1. Introduction

In everyday life, where individuals are frequently faced with changes and unexpected events that demand continuous adjustment ([38]), emotion regulation plays a central role in supporting one’s psychological functioning ([45]; [35]) and well-being ([4]). Although psychological research has traditionally focused on the effectiveness of individual regulation strategies, typically deemed universally adaptive or maladaptive, more recent theoretical models have challenged the generalisability of such an approach, highlighting the “fallacy of uniform efficacy”—the notion that the utility of a given strategy is constant across contexts ([7]). Consequently, regulatory strategies have increasingly been conceptualised in more dynamic terms, with flexibility emerging as a critical factor for adaptive functioning and, ultimately, well-being ([2]; [6]; [7]; [21]; [30]).

Emotion regulation flexibility refers to the ability to dynamically adapt one’s emotional responses by selecting and adjusting strategies according to situational demands and personal goals ([2]). Several accounts ([7]; [15]; [20]; [42]) further highlight three key components of this construct: context sensitivity (i.e., the ability to detect and interpret situational demands; [8]), repertoire breadth (i.e., access to a diverse range of regulation strategies; [14]; [32]), and feedback responsiveness (i.e., the capacity to monitor regulatory outcomes and revise strategies as needed; [5]; [20]). Thus, flexibility is not merely about varying one’s strategy use but about the alignment, or coherence, between regulatory efforts and shifting environmental or internal cues ([2]; [15]).

A substantial body of research has linked emotion dysregulation, conceptualised as difficulty in modulating emotional responses in a functional way or as a rigid reliance on a limited number of strategies, with a heightened vulnerability to anxiety, depression, and related forms of psychological distress ([1]; [19]; [45]). In this framework, flexibility is considered a key component of effective regulation, as limited access to flexible strategies is often interpreted as a core feature of dysregulation ([40]). Moreover, in an empirical study applying latent profile analysis, [9] ([9]) found that deficits in any of the three core components of flexibility, context sensitivity, repertoire, and feedback responsiveness, were associated with higher depressive and anxious symptoms. Context sensitivity emerged as especially relevant for anxious arousal. These insights underscore the importance of having instruments that can directly assess perceived regulatory flexibility while also helping clarify its theoretical overlap with and distinction from related constructs such as dysregulation.

Due to its dynamic and multifaceted nature, emotion regulation flexibility has proven difficult to assess using traditional instruments. In response to this challenge, [34] ([34]) recently developed the Emotion Regulation Flexibility Questionnaire (ERFQ), a self-report instrument designed to assess individuals’ perceived capacity to choose different strategies based on context, modify them when ineffective, and sustain their use for an adequate amount of time. Developed in Australia and grounded in the theoretical model by [2] ([2]), the questionnaire demonstrated excellent internal consistency in its original validation (α = 0.97; [34]). Exploratory factor analysis indicated a clear unidimensional factor structure, with all items loading strongly on a single factor that accounted for 73.9% of the variance. Although concurrent and discriminant validity were not directly assessed in the original validation study due to the lack of comparable measures, theoretically consistent patterns of correlations were observed with related constructs, providing support for the construct validity of the scale ([34]). To the best of our knowledge, the ERFQ represents one of the first systematic attempts to directly measure perceived regulatory flexibility, and no validations have yet been conducted in other languages. Beyond its initial validation, the ERFQ has also been employed in intervention research. [44] ([44]) used the ERFQ in a randomised controlled trial among English-speaking university students in India. They reported high internal consistency of the tool in their sample (α = 0.89) and a significant pre–post increase in flexibility scores in the intervention group, thus supporting the reliability and applied relevance of the measure in a different cultural context from where it was originally developed.

In its original validation, the ERFQ showed that flexibility significantly mediated the relationship between beliefs about emotion controllability and lower levels of stress and anxiety, indicating its potential protective role ([34]). Complementary evidence comes from an intensive longitudinal study, which found that flexibility, particularly strategic variability and context sensitivity, was associated with higher levels of daily positive affect and reduced negative affect ([4]).

The present study aims to examine the psychometric properties of the ERFQ and to adapt it for use in the Italian context, offering a theoretically grounded and empirically tested tool for assessing perceived emotion regulation flexibility. Specifically, we investigate the factorial structure, internal consistency, and measurement invariance of the scale across gender and age groups. Assessing measurement invariance across relevant groups is a crucial psychometric requirement, as it ensures the appropriateness of the instrument for different populations and serves as a prerequisite for making valid and equitable comparisons ([3]; [12], [13]; [49]). Furthermore, we explore the scale’s associations with theoretically relevant constructs, including emotion dysregulation and multidimensional well-being.

By making a validated measure of emotion regulation flexibility available in Italian, this study intends to support both research and applied settings. The ERFQ may offer a valuable tool for identifying individual differences in regulatory flexibility, for examining its associations with other relevant psychological constructs, and for monitoring psychological interventions aimed at fostering adaptive emotional functioning. Ultimately, the present study complements the literature on emotions and emotion regulation by addressing the existing need for reliable and context-sensitive measures of perceived emotion regulation flexibility.

## 2. Materials and Methods

### 2.1. Participants

The study included 887 participants (60.4% female), aged between 18 and 86 years (*M_age_* = 40.0, *SD* = 15.2). Participants reported their demographic characteristics, including marital status, educational attainment, and occupational status. Specifically, 43.4% were single, while 50.1% were married or cohabiting; smaller percentages reported being divorced or widowed (6.0%) or belonging to other categories (0.5%). Regarding education, the largest proportion of participants held a high school diploma (44.6%), followed by a bachelor’s degree (29.0%) and a lower secondary school diploma (22.0%). Fewer participants reported having completed postgraduate education (3.7%), primary education only (0.6%), or other types of qualifications (0.1%). In terms of occupational status, 68.1% of participants were employed, while smaller proportions identified as students (16.3%), retired (6.5%), homemakers (4.9%), unemployed or temporarily unable to work (2.3%), or both students and workers (2.0%).

### 2.2. Procedure

Data were collected across four independent but methodologically similar projects, all aimed at exploring emotion regulation flexibility using the Italian version of the Emotion Regulation Flexibility Questionnaire (IT-ERFQ). Recruitment was conducted through a variety of methods, including the use of social media. Moreover, a snowball sampling strategy was implemented with the support of psychology students, who disseminated a survey link within their personal and professional networks. This strategy helped reach a broader general population sample and may have contributed to reducing the risk of nonresponse bias. For descriptive purposes, the whole dataset was treated as a single sample.

Participants were randomly assigned to two non-overlapping subsamples for exploratory (*N* = 397; 71.4% female; *M_age_* = 40.1, *SD* = 14.9) and confirmatory (*N* = 490; 51.6% female; *M_age_* = 40.0, *SD* = 15.5) factor analyses. Measurement invariance analyses were conducted on the full sample. For analyses involving additional constructs, only participants from data collection waves that administered all relevant measures were included (*N* = 695; 56.8% female; *M_age_* = 40.5, *SD* = 15.4). All participants provided electronic informed consent before taking part in the study. The research adhered to the ethical standards outlined in the Declaration of Helsinki and was approved by the Ethics Committee of the University of Verona (protocol code 2023_29 and date of approval 23 October 2024). Participation was anonymous and voluntary, and participants were informed of their right to withdraw at any time.

### 2.3. Measures

The ERFQ was developed by [34] ([34]) to assess how individuals evaluate their ability to flexibly regulate their emotions. The original version of the ERFQ consists of 10 items, rated on a 7-point Likert scale (1 = “strongly disagree”, 7 = “strongly agree”), with higher scores indicating greater perceived flexibility in emotion regulation. The items reflect different facets of flexibility, including having access to a variety of strategies (e.g., “I use a wide range of different strategies to help regulate my emotions”), adapting strategies to context (e.g., “I use different types of emotion regulation strategies for different types of situations or problems”), and modulating the duration of strategy use (e.g., “When I’m using an emotion regulation strategy, I know the right amount of time to use it for”).

In the present study, the ERFQ was translated and adapted into Italian through a forward- and back-translation procedure involving three fluent Italian researchers and a native English speaker. Prior to the adaptation, we contacted the original authors of the ERFQ and received approval for the adaptation. The Italian version was administered in the present study without item structure or response format changes. The ERFQ was administered online alongside additional self-report measures related to emotion regulation. Participants were instructed to respond based on their general experience with emotion regulation, without reference to specific contexts.

The Difficulties in Emotion Regulation Scale (DERS) was originally developed by [19] ([19]) to assess individuals’ difficulties in regulating negative emotions. In this study, we used the Italian 18-item short version (IT-DERS-SF; [40]), which includes six subscales: nonacceptance of negative emotions (e.g., “When I’m upset, I feel guilty for feeling that way”, Nonacceptance); difficulties engaging in goal-directed behaviours under distress (e.g., “When I’m upset, I have difficulty getting work done”, Goals); impulse control difficulties (e.g., “When I’m upset, I have difficulty controlling my behaviors”, Impulse); limited access to effective emotion regulation strategies (e.g., “When I’m upset, I believe that there is nothing I can do to make myself feel better”, Strategies); lack of emotional awareness (e.g., “I am attentive to my feelings” (reverse item), Awareness), defined as difficulties in attending to and noticing emotional responses; and lack of emotional clarity (e.g., “I have difficulty making sense out of my feelings”, Clarity), referring to difficulties in identifying and understanding one’s emotions. Items are rated on a 5-point Likert scale (1 = “almost never”, 5 = “almost always”), with higher scores indicating greater dysregulation. Subscale scores were computed individually. Following previous recommendations ([40]), an overall dysregulation score was also calculated by averaging all items except those from the Awareness subscale. In the present study (*N* = 695), each dimension reported optimal levels of reliability with α = 0.73 for Nonacceptance, α = 0.86 for Goals, α = 0.88 for Impulse, α = 0.74 for Strategies, α = 0.75 for Awareness, α = 0.75 for Clarity, and α = 0.90 for the total score (excluding Awareness).

Lastly, the Italian version of the Mental Health Continuum—Short Form (MHC–SF; [28]; [36]) was used to measure well-being, conceptualised in three distinct dimensions. The first dimension, emotional well-being, refers to the frequency of positive emotions and overall life satisfaction. Participants were asked how often, during the past month, they had experienced feelings such as being “happy”. The second dimension, psychological well-being, reflects optimal psychological functioning and personal growth (e.g., “Good at managing the responsibilities of your daily life”). The third dimension, social well-being, relates to perceived functioning in one’s social context and community life (e.g., “That you belonged to a community (like a social group, or your neighbourhood)”). In line with Keyes’ model, the MHC–SF integrates both hedonic and eudaimonic conceptions of well-being ([26]; [41]), offering a comprehensive view of positive mental health ([36]). The scale consists of 14 items rated on a 6-point Likert scale from 0 (never) to 5 (every day), with higher scores indicating greater well-being. The Italian MHC–SF has demonstrated good reliability, validity, and measurement invariance across gender ([36]). In the present study (*N* = 687), Cronbach’s α was 0.85 for emotional well-being, 0.85 for psychological well-being, and 0.77 for social well-being.

### 2.4. Data Analysis

A two-sample cross-validation approach was used to validate the Italian version of the ERFQ. Specifically, data from the first subsample (*N* = 397; 71.4% female; *M_age_* = 40.1, *SD* = 14.9) were used to conduct a parallel analysis (PA) and an exploratory factor analysis (EFA), while data from the second subsample (*N* = 490; 51.6% female; *M_age_* = 40.0, *SD* = 15.5) were reserved for confirmatory factor analysis (CFA).

The PA was employed to determine the number of factors to retain by comparing eigenvalues from the observed data with those generated from random correlation matrices ([24]). Factors were retained if their observed eigenvalue exceeded those derived from the simulated data. The EFA was conducted on the first subsample, using the number of factors indicated by the PA. These analyses employed Geomin oblique rotation and maximum likelihood estimation with robust standard errors (MLR), as implemented in Mplus 7.4. Since this study represents the first validation of the instrument in an Italian-speaking population, PA and EFA were conducted to empirically verify the dimensionality of the scale in this new context. This preliminary step ensured that the factor solution was empirically grounded before confirmatory testing, thereby strengthening confidence in the scale’s dimensionality and supporting evidence of its structural stability.

Following the EFA, a preliminary CFA was conducted on the same subsample to obtain model fit indices and inspect modification indices, which are not available in EFA. This step allowed for a more detailed evaluation of model adequacy and the identification of potential sources of misfit. Throughout both the EFA and CFA phases, factor loadings were closely examined to ensure alignment with the intended construct. Items were evaluated for potential issues such as weak loadings or redundancy, which could compromise the clarity of the factor structure. When such issues were identified, problematic items were removed, and the model was re-estimated to assess the impact of their exclusion.

To cross-validate the refined model, a CFA was conducted on the second subsample using Mplus 7.4 and the MLR estimator. To further assess the internal stability of the model, an internal cross-validation procedure was performed: 100 iterations of random resampling were conducted in R using the lavaan package (version 0.6-19; [39]), each involving 70% of the sample drawn without replacement. For each resample, a CFA model was estimated, and factor loadings and fit indices were recorded. The average values across the 100 iterations were used to evaluate the robustness of the model fit and factor loadings.

To evaluate factor models, multiple fit indices were considered, including *χ*^2^, CFI, SRMR, and RMSEA. According to established guidelines, a good model fit is indicated by CFI values ≥ 0.95 (or 0.90–0.95 for reasonable fit) and SRMR/RMSEA values ≤ 0.06 (or 0.06–0.08 for reasonable fit; [31]). The significance of *χ*^2^ was considered but not used as the primary index of fit due to its sensitivity to sample size. As sample size increases, *χ*^2^ is more likely to become significant even when the discrepancies between the observed and estimated covariance matrices are trivial. Therefore, alternative indices were prioritised for evaluating model adequacy ([22]).

The measurement invariance of the final model was tested across gender and age groups using the multiple-group approach ([33]; [37]; [49]). Analyses were conducted in Mplus 7.4 using the MLR estimator. Configural, metric, and scalar invariances were examined sequentially following current best practices. In each case, model fit was evaluated using conventional fit indices (i.e., CFI ≥ 0.95; RMSEA/SRMR ≤ 0.06 for good fit), and comparisons across models were based on changes in χ^2^ ([43]), as well as on relative differences in CFI, RMSEA, and SRMR. Invariance was tested across gender (male vs. female) and two age groups (up to 40 years vs. 41 years and older), based on the distribution of the sample. The analysis also included tests of latent mean equality.

To evaluate convergent validity, bivariate correlations were computed between IT-ERFQ scores and dimensions of the IT-DERS-SF. The DERS was selected as a theoretically relevant comparator, as emotion regulation flexibility involves the ability to select and adjust strategies based on situational demands, monitor their effectiveness, and remain attuned to one’s emotional states. Accordingly, we expected negative correlations between IT-ERFQ scores and all IT-DERS-SF subscales, with particularly strong associations for those representing the theoretical counterpart of emotion regulation flexibility—namely, Strategies, Awareness, and Clarity. Correlations were corrected for attenuation based on Cronbach’s α values for both scales ([46]), to account for measurement error and better estimate the true association between constructs.

To further strengthen the evaluation of construct validity, the correlation analyses were extended with additional formal validity testing procedures. Specifically, the Average Variance Extracted (AVE) was computed for the IT-ERFQ. It represents an index of the proportion of variance captured by the construct relative to measurement error. AVE values greater 0.50 are generally considered acceptable and indicative of convergent validity ([16]). The square root of the AVE was then compared with the correlations between the IT-ERFQ and the IT-DERS-SF subdimensions to assess discriminant validity. Discriminant validity at the construct level is established when the square root of the AVE exceeds the highest correlation with any other latent variable ([16]). In addition, the Heterotrait–Monotrait ratio (HTMT ratio; [23]) was computed between the IT-ERFQ and the total IT-DERS-SF score. Given that only two constructs were considered for this analysis, the HTMT ratio was calculated as the average standardised correlation between IT-ERFQ and IT-DERS-SF items, normalised by the average within-construct item correlations. For this analysis, a threshold of 0.85 is frequently cited as indicative of adequate discriminant validity ([10]; [29]), although a less conservative threshold of 0.90 is also considered acceptable ([17]; [48]).

Finally, to further examine the external validity of the scale, correlations between IT-ERFQ scores and dimensions of well-being, as measured by the MHC–SF, were computed. Given that emotion regulation flexibility involves adaptively managing emotions in response to situational demands, and consistent with previous evidence associating effective emotion regulation with greater well-being, we anticipated positive correlations between IT-ERFQ scores and emotional, psychological, and social well-being dimensions measured by the MHC–SF. Correlations were again corrected for attenuation based on the Cronbach’s α of both scales.

## 3. Results

A parallel analysis (Figure 1) conducted on the first subsample (*N* = 397) supported a unidimensional structure, as only the first observed eigenvalue exceeded those derived from random correlation matrices, thus confirming the original unidimensional structure of the ERFQ proposed by [34] ([34]).

The subsequent exploratory factor analysis confirmed that most items loaded substantially (ranging from 0.694 to 0.809) on a single factor. Nevertheless, several items exhibited relatively high residual variances (>0.45), including Items 1 (0.483), 2 (0.518), 8 (0.518), and 9 (0.491), indicating that a non-negligible portion of their variance was not captured by the common factor. These patterns did not warrant immediate exclusion but highlighted the need for further examination.

The model was subsequently assessed in a preliminary CFA using the same sample. The initial 10-item CFA model indicated a suboptimal fit (χ^2^(45) = 1601.81, *p* < 0.001; RMSEA = 0.092, 90% CI [0.078, 0.108]; CFI = 0.924; SRMR = 0.045), with acceptable CFI and SRMR values but poor RMSEA, suggesting a less-than-optimal model fit ([25]; [31]). Modification indices (MIs) and theoretical considerations pointed to significant redundancy involving Item 1 and Item 8, which showed correlated residual variances with multiple other items. These items were removed, and a revised eight-item model was tested, resulting in a substantially improved fit (χ^2^(28) = 1152.75, *p* < 0.001; RMSEA = 0.072, 90% CI [0.052, 0.093]; CFI = 0.964; SRMR = 0.033). This refined structure was retained for cross-validation. A confirmatory factor analysis of the eight-item model was then conducted on the second subsample (N = 490) using the MLR estimator in Mplus 7.4. The model showed an acceptable fit to the data (χ^2^(20) = 81.76, *p* < 0.001; RMSEA = 0.079, 90% CI [0.062, 0.098]; CFI = 0.938; SRMR = 0.041). To evaluate the internal stability of the model, a resampling procedure was implemented: 100 iterations of the CFA were conducted on randomly drawn subsamples (70% of the data without replacement). The average fit indices across these replications indicated an overall acceptable model fit (mean χ^2^ = 108.63; mean RMSEA = 0.079; mean CFI = 0.939; mean SRMR = 0.041), supporting the robustness of the solution. The standardised factor loadings and corresponding squared multiple correlations (*R*^2^) for the final eight-item model are reported in Table 1 (means of standardised factor loadings across replications for Items 2, 3, 4, 5, 6, 7, 9, and 10 were 0.60, 0.59, 0.61, 0.59, 0.59, 0.59, 0.62, and 0.61, respectively).

The measurement invariance of the eight-item model was tested using the MLR estimator by estimating configural, metric, and scalar models across gender (520 females, 334 males) and age groups (477 participants aged ≤ 40 years; 379 aged ≥ 41 years). The configural model demonstrated acceptable fit across gender (χ^2^(40) = 158.01, *p* < 0.001; RMSEA = 0.083; CFI = 0.942; SRMR = 0.040) and age (χ^2^(40) = 147.49, *p* < 0.001; RMSEA = 0.079; CFI = 0.947; SRMR = 0.038). Full metric and scalar invariances were supported in both comparisons, as indicated by non-significant χ^2^ differences and negligible changes in fit indices across nested models (Table 2). No significant differences in latent means were observed across gender. However, comparison of latent means across age groups revealed a statistically significant difference (Δχ^2^(1) = 4.88, *p* = 0.027), with older participants showing higher levels of perceived emotion regulation flexibility. The effect size of this difference was small (χ^2^-based ES = 0.08), with a latent mean difference of 0.18 (SE = 0.08, *p* = 0.029) corresponding to a Cohen’s *d* of 0.16. This value falls within the range typically interpreted as small ([11]), suggesting that although statistically significant, the magnitude of this age-related difference is modest.

The internal consistency of the IT-ERFQ was assessed using Cronbach’s α and Composite Reliability (CR). The 10-item version of the scale demonstrated excellent reliability (α = 0.93). The shortened eight-item version also showed very good internal consistency (α = 0.92), indicating that the reduction in item number did not compromise reliability. The Composite Reliability for the eight-item model, calculated with the CFA sample (*N* = 490) using the MLR estimator, was 0.89, further supporting the internal coherence of the final scale.

The correlations between IT-ERFQ scores and IT-DERS-SF subscales were all negative and statistically significant (see Table 3), supporting the convergent validity of the IT-ERFQ.

In line with theoretical expectations, [47]’s ([47]) test indicated that the strongest associations were observed for Strategies (*r* = −0.35), Awareness (*r* = −0.34), and Clarity (*r* = −0.29), which reflect core dimensions that are conceptually opposed to regulatory flexibility. Weaker, yet still significant, correlations were found with Goals (*r* = −0.22), Impulse (*r* = −0.21), and Nonacceptance (*r* = −0.19). The aggregate IT-DERS-SF score (excluding Awareness) also showed a moderate negative correlation with the IT-ERFQ (*r* = −0.30). These results are consistent with the notion that individuals who perceive themselves as more flexible in regulating emotions report fewer difficulties across multiple aspects of emotion regulation.

The convergent validity of the IT-ERFQ was supported by an AVE of 0.50, meeting the recommended threshold of 0.50 ([16]). The square root of the AVE (0.71) exceeded the correlations between the IT-ERFQ and all IT-DERS-SF subdimensions, indicating adequate discriminant validity. The HTMT ratio between the IT-ERFQ and the total IT-DERS-SF score (the average standardised correlation between IT-ERFQ and IT-DERS-SF items, normalised by the average within-construct item correlation) was 0.65, below the conservative 0.85 threshold ([23]), further supporting discriminant validity.

The correlations between IT-ERFQ scores and the Mental Health Continuum dimensions were positive and statistically significant (see Table 4).

The IT-ERFQ was positively associated with all three well-being dimensions, with the strongest correlation observed for psychological well-being (*r* = 0.37), followed by social (*r* = 0.31) and emotional well-being (*r* = 0.28). Steiger’s Z tests confirmed that the association with psychological well-being was significantly stronger than those with social (Z_H_ = 2.05, *p* = 0.040) and emotional well-being (Z_H_ = 3.28, *p* = 0.001). These results support the external validity of the IT-ERFQ and are consistent with the idea that individuals who perceive themselves as more flexible in regulating emotions tend to report higher levels of perceived psychological functioning and well-being.

## 4. Discussion

The present study aimed to address a gap in the literature by adapting and examining the psychometric properties of the Emotion Regulation Flexibility Questionnaire (ERFQ; [34]) within the Italian context. In addition to the validation of the original version, the study included a scale refinement phase, which led to an 8-item Italian version, compared to the 10-item Australian version. This reduction was driven by theoretical considerations and supported by statistical evidence, with the aim of enhancing conceptual clarity and efficiency, thereby facilitating the scale’s application in both applied and research settings.

Exploratory and confirmatory factor analyses demonstrated a good model fit for the reduced eight-item version, confirming the hypothesised unidimensional structure. Internal reliability indices, including Cronbach’s alpha and Composite Reliability, supported the internal coherence of the selected items, suggesting that the reduction process preserved a solid representation of the construct and contributed to strengthening its internal consistency. Measurement invariance analyses confirmed the structural stability of the scale across gender and two age groups (≤40 years vs. >40 years), supporting the meaningful comparability of scores between subgroups. Establishing measurement invariance is particularly important, as it ensures that the observed differences reflect substantive distinctions rather than measurement artefacts. In this regard, previous evidence has highlighted gender differences in key components of emotion regulation flexibility, such as context sensitivity and repertoire, underscoring the importance of considering gender when examining ER processes ([18]), while findings on age differences remain inconclusive. As [27] ([27]) notes, the idea that older adults are better at emotion regulation has emerged as an intuitively appealing explanation for their relatively high levels of affective well-being despite other age-related declines, yet current evidence does not clearly support this claim. Future research will be needed to clarify these age-related dynamics, and brief, psychometrically robust measures such as the present one may help advance this line of inquiry. In our sample, a significant difference in latent means was found, with higher scores among participants over 40 years old, suggesting that this group tends to perceive themselves as more flexible in regulating emotions. Although the cross-sectional design does not allow for inferences about age-related developmental trends, this finding aligns with the hypothesis that life experience may foster greater emotional awareness and adaptability in emotion regulation.

As for convergent validity, negative correlations between IT-ERFQ scores and the Difficulties in Emotion Regulation Scale (IT-DERS-SF) confirmed that higher regulatory flexibility is associated with lower perceived emotion dysregulation. In line with our hypotheses, based on the theoretical models by [2] ([2]) and [7] ([7]), stronger associations emerged with the IT-DERS-SF subscales Strategies, Clarity, and Awareness, which are theoretically more closely related to the construct of flexibility. These findings are consistent with the view that flexibility requires, as foundational components, perceived access to effective strategies, emotional awareness, and attentiveness to one’s emotional states ([2]; [7]). By contrast, the IT-DERS-SF dimensions least associated with the IT-ERFQ, namely Goals, Impulse, and Nonacceptance, appear to reflect more reactive and behavioural aspects of dysregulation, such as difficulty inhibiting intense emotional responses or tolerating negative affective states ([19]). These results strengthen the theoretical validity of the scale and underscore its practical relevance, suggesting that the ERFQ may represent a useful tool for identifying specific difficulties in emotion regulation, with potential applications in research, clinical assessment, and intervention design.

Regarding the relationship between perceived flexibility and perceived well-being, IT-ERFQ scores were positively associated with all dimensions of well-being measured by the MHC-SF, with the strongest association emerging for psychological well-being. This finding is consistent with previous evidence ([4]; [9]; [21]; [34]) and also suggests that perceived emotion regulation flexibility may be associated with broader aspects of psychological and social well-being, beyond the mere absence of distress. In particular, the stronger correlation with psychological well-being may reflect metacognitive processes involved in regulatory flexibility, which entails awareness, intentionality, and the capacity for strategic adaptation. These qualities appear especially relevant for supporting a positive self-view, a sense of internal coherence, perceived personal growth, and environmental mastery—core components of psychological well-being as conceptualised in the MHC-SF ([36]). Furthermore, the positive associations observed with social well-being suggest that regulatory flexibility may contribute to adaptive functioning across multiple domains, including individuals’ perceived sense of belonging and social contribution. These results offer preliminary support for the hypothesis that flexible regulation processes are not only protective against emotional distress but also instrumental in sustaining eudaimonic functioning. Future research using longitudinal and multimethod approaches is needed to clarify the directionality and underlying mechanisms of these associations. Ecological momentary assessment (EMA), for instance, has been proposed as a promising method for capturing the dynamic nature of emotion regulation flexibility in daily life ([15]). Although it presents unique methodological challenges, EMA could complement self-report measures such as the IT-ERFQ by providing fine-grained data on regulatory processes as they unfold in context. Similarly, the use of experience sampling methodologies (ESMs) has been recommended to capture emotion regulation flexibility across everyday situations, as part of broader efforts to advance theoretical models of this construct ([42]).

Taken together, the results of the present study highlight the potential of the IT-ERFQ as a tool for capturing individuals’ perceived ability to flexibly regulate emotions and support its use in both basic and applied research. Future research could further investigate the role of regulatory flexibility in relation to additional constructs, as well as its impact across different developmental stages. In theoretical and applied contexts, the IT-ERFQ may offer a promising tool for identifying individual differences in emotion regulation tendencies and informing intervention strategies.

## 5. Conclusions

The present study provides a novel contribution to the understanding of emotion regulation flexibility and lays the basis for future research and practice. While the cross-sectional design and exclusive reliance on self-report measures limit causal interpretations and may be subject to response biases, this design was deemed appropriate for a first investigation of the ERFQ in the Italian context. Future research should examine the IT-ERFQ’s predictive utility and explore its applicability in clinical populations and diverse cultural contexts.

## Figures and Tables

**Figure 1 ejihpe-15-00165-f001:**
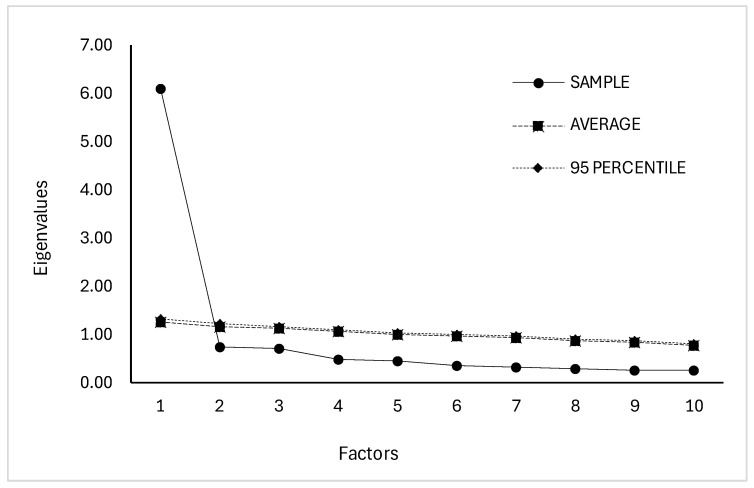
Parallel analysis. Scree plot of real data eigenvalues and average random data eigenvalues.

**Table 1 ejihpe-15-00165-t001:** Standardised factor loadings and squared multiple correlations (R^2^) from the final 8-item CFA model.

Item	*M*	*SD*	Item Total Correlation	Standardised λ	SE	R^2^
2	4.11	1.63	0.69	0.74	0.03	0.54
3	3.97	1.70	0.72	0.77	0.03	0.59
4	4.62	1.63	0.62	0.67	0.04	0.44
5	4.44	1.67	0.70	0.76	0.03	0.57
6	4.00	1.64	0.69	0.75	0.03	0.56
7	3.94	1.65	0.71	0.75	0.03	0.57
9	3.91	1.56	0.53	0.56	0.04	0.31
10	4.48	1.60	0.64	0.67	0.03	0.45

*Note*: estimates are based on CFA using the MLR estimator (*N* = 490; *M* = 4.18, *SD* = 1.23).

**Table 2 ejihpe-15-00165-t002:** Fit indices and model comparisons for measurement invariance across gender and age groups.

Group	Model	χ^2^ (df)	*p*	RMSEA	CFI	SRMR	ΔS-Bχ^2^ (df)	*p*	ΔRMSEA	ΔCFI	ΔSRMR
Gender	Configural	158.01(40)	<0.001	0.083	0.942	0.040	-	-	-	-	-
	Metric	167.47(47)	<0.001	0.077	0.940	0.042	2.13(7)	0.952	0.006	0.002	−0.002
	Scalar	177.85(54)	<0.001	0.073	0.939	0.042	3.07(7)	0.878	0.004	0.001	0.000
	Latent Means	178.95(55)	<0.001	0.073	0.939	0.042	0.01(1)	0.926	0.000	0.000	0.000
Age	Configural	147.49(40)	<0.001	0.079	0.947	0.038	-	-	-	-	-
	Metric	161.01(47)	<0.001	0.075	0.944	0.045	6.91(7)	0.439	0.004	0.003	−0.007
	Scalar	178.40(54)	<0.001	0.073	0.939	0.047	14.01(7)	0.051	0.002	0.005	−0.002
	Latent Means	182.76(55)	<0.001	0.074	0.937	0.054	4.88(1)	0.027	−0.001	0.002	−0.007

*Note*: Estimates are based on multiple-group CFA using the MLR estimator in Mplus 7.4. ΔS-Bχ^2^ = Satorra–Bentler scaled chi-square difference test.

**Table 3 ejihpe-15-00165-t003:** Descriptive statistics and correlations between IT-ERFQ and IT-DERS-SF subscales (corrected for attenuation).

IT-DERS-SF	M	SD	Correlation with IT-ERFQ
Awareness	2.44	0.95	−0.34 ***
Clarity	2.10	0.91	−0.29 ***
Goals	2.68	1.03	−0.22 ***
Impulse	2.03	1.01	−0.21 ***
Nonacceptance	2.12	0.93	−0.19 ***
Strategies	2.19	0.91	−0.35 ***
Total score	2.26	0.66	−0.30 ***

*Note:* Higher scores on the IT-ERFQ reflect greater perceived emotion regulation flexibility; higher scores on the DERS-SF indicate greater emotion dysregulation. *** *p* < 0.001.

**Table 4 ejihpe-15-00165-t004:** Descriptive statistics and correlations between IT-ERFQ and MHC-SF subscales (corrected for attenuation).

MHC-SF	M	SD	IT-ERFQ
Emotional well-being	3.24	1.07	0.28 ***
Social well-being	1.91	1.05	0.31 ***
Psychological well-being	3.24	1.05	0.37 ***

*Note*: Higher scores on the ERFQ reflect greater perceived emotion regulation flexibility; higher scores on the MHC-SF indicate greater well-being. *** *p* < 0.001.

## Data Availability

The raw data supporting the conclusions of this article will be made available by the authors on request. The data are not publicly available due to privacy concerns.

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
