# Peer review of "Investigating the Psychometric Properties of the Emotion Regulation Flexibility Questionnaire in the Italian Context"

_ejihpe, 2025, doi:10.3390/ejihpe15080165_

Round 1
Reviewer 1 Report
Comments and Suggestions for Authors
The manuscript presents a methodologically rigorous and well-executed analysis, including alternative CFA models, measurement invariance testing, and internal stability checks through resampling procedures. These elements considerably strengthen the evidence for the structural validity of the instrument. To further enhance the psychometric evaluation, I recommend incorporating formal validity testing procedures beyond the correlation matrix. In particular, the inclusion of Average Variance Extracted (AVE) for assessing convergent validity, along with the Fornell-Larcker criterion and Heterotrait-Monotrait Ratio (HTMT) for assessing discriminant validity, would provide a more comprehensive and robust validation framework. These additions would reinforce the value of the instrument for future applications.
Additionally, I suggest the authors conduct a more extensive and up-to-date literature review, integrating recent studies from the past five years. This would strengthen the theoretical grounding of the manuscript, demonstrate its relevance within the current research landscape, and enrich the discussion with contemporary insights.
Finally, I recommend that the authors explicitly justify the change in estimator between analyses (e.g., from MLR to MLMV), as such a shift may affect the comparability of model fit statistics or chi-square difference testing. Clarifying the rationale for this methodological choice would further support the transparency and replicability of the analyses.
Author Response
Authors’ Response to Reviewers’ Comments
Reviewer’s Comments (RC)
Authors’ Response (AR)
---------------------------------------------------------------------------------------
Reviewer 1
RC1: The manuscript presents a methodologically rigorous and well-executed analysis, including alternative CFA models, measurement invariance testing, and internal stability checks through resampling procedures. These elements considerably strengthen the evidence for the structural validity of the instrument.
AR1: We are glad you appreciate our work on the manuscript. Thank you very much for your constructive and supportive comments which helped us to substantially revise it.
RC2: To further enhance the psychometric evaluation, I recommend incorporating formal validity testing procedures beyond the correlation matrix. In particular, the inclusion of Average Variance Extracted (AVE) for assessing convergent validity, along with the Fornell-Larcker criterion and Heterotrait-Monotrait Ratio (HTMT) for assessing discriminant validity, would provide a more comprehensive and robust validation framework. These additions would reinforce the value of the instrument for future applications.
AR2: Thank you for this valuable suggestion. In the new version of the manuscript, we extended our evaluation of construct validity by incorporating the formal validity testing procedures recommended by the Reviewer. Specifically, we computed the Average Variance Extracted (AVE) for the ERFQ, which reached 0.50, meeting the commonly accepted threshold for convergent validity (Fornell & Larcker, 1981).
To assess discriminant validity, we compared the square root of the AVE (0.71) with the correlations between IT-ERFQ and all IT-DERS-SF subdimensions. The square root of the AVE exceeded these correlations, indicating adequate discriminant validity in line with the Fornell-Larcker criterion. Additionally, we calculated the Heterotrait-Monotrait Ratio (HTMT; Henseler et al., 2015) between ERFQ and the total IT-DERS-SF score. The HTMT ratio was 0.65, well below the conservative .85 threshold, further supporting discriminant validity.
These additional analyses strengthen the psychometric evaluation and confirm that the IT-ERFQ demonstrates both convergent and discriminant validity, thereby reinforcing its suitability for future applications.
To further strengthen the evaluation of construct validity, the correlation analyses were extended with additional formal validity testing procedures. Specifically, the Average Variance Extracted (AVE) was computed for the IT-ERFQ. It represents an index of the proportion of variance captured by the construct relative to measurement error. AVE values greater 0.50 are generally considered acceptable and indicative of convergent validity (Fornell & Larcker, 1981). The square root of the AVE was then compared with the correlations between IT-ERFQ and the IT-DERS-SF subdimensions to assess discriminant validity. Discriminant validity at the construct level is established when the square root of the AVE exceeds the highest correlation with any other latent variable (Fornell & Larcker, 1981). In addition, the Heterotrait-Monotrait Ratio (HTMT; Henseler et al., 2015) was computed between IT-ERFQ and the total IT-DERS-SF score. Given that only two constructs were considered for this analysis, HTMT was calculated as the average standardized correlation between IT-ERFQ and IT-DERS-SF items, normalized by the average within-construct item correlations. For this analysis, a threshold of 0.85 is frequently cited as indicative of adequate discriminant validity (Clark & Watson, 1995; Kline, 2011), although a less conservative threshold of 0.90 is also considered acceptable (Gold et al., 2001; Teo et al., 2008).
(Lines 267–283)
Convergent validity of IT-ERFQ was supported by an AVE of .50, meeting the recommended threshold of .50 (Fornell & Larcker, 1981). The square root of AVE (.71) exceeded the correlations between IT-ERFQ and all IT-DERS-SF subdimensions, indicating adequate discriminant validity. The HTMT ratio between IT-ERFQ and the total IT-DERS-SF score (the average standardized correlation between IT-ERFQ and IT-DERS-SF items, normalized by the average within-construct item correlation) was .65, below the conservative .85 threshold (Henseler et al., 2015), further supporting discriminant validity.
(Lines 372–378)
RC3: Additionally, I suggest the authors conduct a more extensive and up-to-date literature review, integrating recent studies from the past five years. This would strengthen the theoretical grounding of the manuscript, demonstrate its relevance within the current research landscape, and enrich the discussion with contemporary insights.
AR3: We thank the Reviewer for this valuable suggestion. In the revised manuscript, we have expanded the literature review to strengthen the theoretical foundation of the study and ensure alignment with recent advances in the field. In particular, we integrated contributions from the past five years that further develop the conceptualization and application of emotion regulation flexibility. For example, we cited Chen and Bonanno (2021) to provide empirical support for the multicomponent model of flexibility, Sharma and Singh (2024) to highlight the applied relevance of the ERFQ in intervention research, and Sanchez-Lopez (2021) together with English and Eldesouky (2020) to enrich the discussion with contemporary conceptual and methodological insights.
These updates were incorporated mostly in the Introduction and Discussion, and they are highlighted in light blue in the revised manuscript.
RC4: Finally, I recommend that the authors explicitly justify the change in estimator between analyses (e.g., from MLR to MLMV), as such a shift may affect the comparability of model fit statistics or chi-square difference testing. Clarifying the rationale for this methodological choice would further support the transparency and replicability of the analyses.
AR4: We appreciate the reviewer’s insightful comment regarding the change of estimator between analyses. Initially, we employed MLMV because it automatically provides the chi-square difference test (Δχ²). However, we fully agree that maintaining consistency in the estimator enhances comparability and transparency. In the revised manuscript, we have updated the invariance analyses to use MLR, ensuring alignment across all analyses and enabling the use of the appropriate chi-square difference testing (e.g., Satorra–Bentler scaled chi-square difference test). The overall results remained substantively unchanged; however, the values reported in Table 2 have been updated accordingly.
-----------------------------------------------------------------------------------
Reviewer 2
RC1: Please indicate a gender distribution in the abstract.
AR1: Thank you, we have added the gender distribution in the abstract.
RC2: Remove a reference from the abstract.
AR2: Thanks, we have deleted the reference from the abstract.
RC3: In the introduction, please describe the psychometric performance of the validated scale in different language versions. Also please include a piece of information about its factor structure. Please review its performance.
AR3: We thank the reviewer for this helpful comment. In the revised manuscript, we have expanded the Introduction to better address the psychometric performance of the ERFQ. Specifically, we now note that, to the best of our knowledge, the scale has not yet been validated in other languages. We describe its original psychometric properties, reporting that exploratory factor analysis supported a unidimensional factor structure, with all items loading strongly on a single factor that accounted for 73.9% of the variance, and that the total score demonstrated excellent internal consistency (α = .97; Monsoon et al., 2022). We also included a recent study that employed the original English version of the ERFQ in a different cultural context (Sharma & Singh, 2024), which reported high internal consistency (α = .89) and significant pre–post improvements in flexibility scores following an intervention. These additions provide a clearer overview of the scale’s psychometric properties and its applied relevance.
Developed in Australia and grounded in the theoretical model by Aldao et al. (2015), the questionnaire demonstrated excellent internal consistency in its original validation (α = .97; Monsoon et al., 2022). Exploratory factor analysis indicated a clear unidimensional factor structure, with all items loading strongly on a single factor that accounted for 73.9% of the variance. Although concurrent and discriminant validity were not directly assessed in the original validation study due to the lack of comparable measures, theoretically consistent patterns of correlations were observed with related constructs, providing support for the construct validity of the scale (Monsoon et al., 2022). To the best of our knowledge, the ERFQ represents one of the first systematic attempts to directly measure perceived regulatory flexibility, and no validations have yet been conducted in other languages. Beyond its initial validation, the ERFQ has also been employed in inter-vention research. Sharma and Singh (2024) used the ERFQ in a randomized controlled trial among English-speaking university students in India. They reported high internal consistency of the tool in their sample (α = .89) and a significant pre–post increase in flexibility scores in the intervention group, thus supporting the reliability and applied relevance of the measure in a different cultural context from where it was originally developed.
(Lines 75–91)
RC4: Please cite all papers which use the same data presented here. Please do not blind the university, as even the authors are visible for reviewers in this journal.
AR4: We thank the Reviewer for this comment. At present, no publications have been based on the data used for the current validation study. We have taken off the blind option.
RC5: Please present full descriptive statistics at an item level and total scores for the validated scale. Total scores are also needed for other scales used here.
AR5: We thank the reviewer for this helpful suggestion. In the revised manuscript, Table 1 has been updated to include, for each item of the validated scale, the mean (M), standard deviation (SD), and item–total correlation. Additionally, a table note now reports the overall mean and SD for the total scale score. For the remaining constructs, descriptive statistics for the total scores (M and SD) are reported in Tables 3 and 4.
RC6: Age and gender differences are good, but please indicate effect sizes.
AR6: We thank the reviewer for this comment. In the revised manuscript, we clarify that no significant gender differences were observed. For age, which showed a significant difference, we computed the effect size, which was found to be small. This information has now been explicitly added to the Results section of the revised version as follows:
No significant differences in latent means were observed across gender. However, comparison of latent means across age groups revealed a statistically significant difference (Δχ²(1) = 4.88, p = .027), with older participants showing higher levels of perceived emotion regulation flexibility. The effect size of this difference was small (χ²-based ES = 0.08), with a latent mean difference of 0.18 (SE = 0.08, p = .029) corresponding to a Cohen’s d of 0.16, which falls within the range typically interpreted as small (Cohen, 1988), suggesting that although statistically significant, the magnitude of this age-related difference is modest.
(Lines 336 – 343)
RC7: Please describe rationale of EFA testing while the scale has a known factor structure. Overall, an EFA is not necessary.
AR7: We thank the reviewer for this observation. In the revised manuscript, we clarified that, since this study represents the first validation of the instrument in an Italian-speaking population, we conducted PA and EFA to empirically verify its dimensionality before confirmatory testing. In the revised manuscript, this has been clarified in the analysis strategy section as follows
Since this study represents the first validation of the instrument in an Italian-speaking population, PA and EFA were conducted to empirically verify the dimensionality of the scale in this new context. This preliminary step ensured that the factor solution was empirically grounded before confirmatory testing, thereby strengthening confidence in the scale’s dimensionality and supporting evidence of its structural stability.
(Lines 218 – 222)
RC8: Please indicate versions of MPlus, R, lavaan etc. Please cite references for MI testing.
AR8: We thank the reviewer for this helpful comment. In the revised manuscript, we have added the appropriate references for measurement invariance testing and specified the software versions used (MPlus, R, and lavaan).
R using the lavaan package (version 0.6-19; Rosseel, 2012), (Line 234)
Reference for MI testing:
(Meredith, 1993; Putnick & Bornstein,2016; Vandenberg & Lance 2000). (Lines 248-249)
Analyses were conducted in Mplus 7.4 (Line 249)
RC9: It seems that different estimators were used in CFA and MI. Please clarify and justify.
AR9: We appreciate the reviewer’s insightful comment regarding the change of estimator between analyses. Initially, we employed MLMV because it automatically provides the chi-square difference test (Δχ²). However, we fully agree that maintaining consistency in the estimator enhances comparability and transparency. In the revised manuscript, we have updated the invariance analyses to use MLR, ensuring alignment across all analyses and enabling the use of the appropriate chi-square difference testing (e.g., Satorra–Bentler scaled chi-square difference test). The overall results remained substantively unchanged; however, the values reported in Table 2 have been updated accordingly.
RC10: Overall, deleting items in the scale strongly reduces its cross-cultural applicability. I do not find any significant reasons to delete items. Suboptimal fit as suggested by CFA, perchance, if you cite rigorous conventions. Regarding optimal and suboptimal fit, please refer to Bagby, R. M., Taylor, G. J., Quilty, L. C., & Parker, J. D. (2007). Reexamining the factor structure of the 20-item Toronto alexithymia scale: commentary on Gignac, Palmer, and Stough. Journal of Personality Assessment, 89(3), 258–264. https://doi.org/10.1080/00223890701629771
AR10: We thank the reviewer for this comment and for drawing our attention to Bagby et al. (2007). We fully agree with their caution that removing items from well-established psychometric scales only to improve statistical model fit may compromise construct validity and cross-cultural applicability. However, regarding the ERFQ scale validated in our study, the removal of two items was based on specific considerations. Our study represents one of the first adaptations of this scale in a non-English-speaking population. To the best of our knowledge, the ERFQ had previously only been tested in English, both in its original development in Australia and in subsequent research conducted in India, and therefore weak items had not been identified in prior research. During our validation process, we found that, consistent with the original version, the scale was best represented as unidimensional, but two items showed problematic behaviour as they were highly correlated with each other and displayed weak loadings on the underlying construct, and also presented a degree of semantic redundancy in the Italian translation. The decision to remove these items was therefore not made simply to improve model fit, but to strengthen internal consistency and construct validity in the Italian adaptation. Moreover, analyses of convergent, discriminant, and external validity confirmed that the shortened version without these two items performs adequately and preserves the intended measurement properties.
RC11: Please present separate conclusions.
AR11: Thank you for your indication. We have revised the closing paragraph and separated it as a conclusion section.
The present study provides a novel contribution to the understanding of emotion regulation flexibility and lays the basis for future research and practice. While the cross-sectional design and exclusive reliance on self-report measures limit causal interpretations and may be subject to response biases, this design was deemed appropriate for a first investigation of the ERFQ in the Italian context. Future research should examine the IT-ERFQ’s predictive utility and explore its applicability in clinical populations and diverse cultural contexts.
(Lines 479-484)
Reviewer 2 Report
Comments and Suggestions for Authors
Please find the comments:
- Please indicate a gender distribution in the abstract.
- Remove a reference from the abstract.
- In the introduction, please describe the psychometric performance of the validated scale in different language versions. Also please include a piece of information about its factor structure. Please review its performance.
- Please cite all papers which use the same data presented here. Please do not blind the university, as even the authors are visible for reviewers in this journal.
- Please present full descriptive statistics at an item level and total scores for the validated scale. Total scores are also needed for other scales used here.
- Age and gender differences are good, but please indicate effect sizes.
- Please describe rationale of EFA testing while the scale has a known factor structure. Overall, an EFA is not necessary.
- Please indicate versions of MPlus, R, lavaan etc.
- Please cite references for MI testing.
- It seems that different estimators were used in CFA and MI. Please clarify and justify.
- Overall, deleting items in the scale strongly reduces its cross-cultural applicability. I do not find any significant reasons to delete items. Suboptimal fit as suggested by CFA, perchance, if you cite rigorous conventions. Regarding optimal and suboptimal fit, please refer to Bagby, R. M., Taylor, G. J., Quilty, L. C., & Parker, J. D. (2007). Reexamining the factor structure of the 20-item Toronto alexithymia scale: commentary on Gignac, Palmer, and Stough. Journal of Personality Assessment, 89(3), 258–264. https://doi.org/10.1080/00223890701629771
- Please present separate conclusions.
Author Response

(The authors gave the same response as above.)

Round 2
Reviewer 2 Report
Comments and Suggestions for Authors
Thank you for your work on the revised version of the paper. I deem it has been improved in a significant way. Congratulations!
These are minor comments which should be addressed:
- Please use zeros before decimal places in numbers as it is in accordance with the journal's requirements.
- Please indicate whether approval from the authors of the English version of the ERFQ was received before adaptation procedures.
- Please write SD, not sd.
Good luck!
Author Response
Comment 1: Please use zeros before decimal places in numbers as it is in accordance with the journal’s requirements.
Response 1: We thank the reviewer for this note. All decimal numbers have been revised accordingly. These changes have been applied consistently throughout the manuscript and tables and have been highlighted in light blue for ease of reference.
Comment 2: Please indicate whether approval from the authors of the English version of the ERFQ was received before adaptation procedures.
Response 2: We thank the reviewer for this important point. Prior to the adaptation, we contacted the original authors of the ERFQ and received approval for the adaptation. This information has been added to the Methods section [lines 163–164] and highlighted in blue.
Comment 3: Please write SD, not sd.
Response 3: We thank the reviewer for this note. All occurrences of sd have been corrected to SD in the manuscript and tables. These changes have also been highlighted in light blue for ease of reference.